# The Mental Health Impacts of a Pandemic: A Multiaxial Conceptual Model for COVID-19

**DOI:** 10.3390/bs13110912

**Published:** 2023-11-08

**Authors:** David Dias Neto, Ana Nunes da Silva

**Affiliations:** 1School of Psychology, ISPA—Instituto Universitário, 1140-041 Lisbon, Portugal; 2Applied Psychology Research Center Capabilities & Inclusion, ISPA—Instituto Universitário, 1149-041 Lisbon, Portugal; 3Faculdade de Psicologia, Universidade de Lisboa, 1649-004 Lisbon, Portugal; acsilva@psicologia.ulisboa.pt

**Keywords:** COVID-19, pandemic, mental health, public health

## Abstract

The COVID-19 pandemic substantially impacted the mental health of the general population and particularly vulnerable individuals and groups. A wealth of research allows for estimating this impact and identifying relevant factors contributing to or mitigating it. The current paper presents and synthesizes this evidence into a multiaxial model of COVID-19 mental health impacts. Based on existing research, we propose four axes: (1) Exposure to COVID-related events; (2) Personal and social vulnerability, such as previous mental health problems or belonging to a vulnerable group; (3) Time, which accounts for the differential impacts throughout the development of the pandemic; and (4) Context, including healthcare and public policies, and social representations of the illness influencing individual emotional reactions and relevant behaviors. These axes help acknowledge the complexity of communities’ reactions and are pragmatic in identifying and prioritizing factors. The axes can provide individual information (i.e., more exposure is harmful) and account for interactions (e.g., exposure in an early phase of the pandemic differs from a later stage). This model contributes to the reflections of the evidence and informs the mental health response to the next pandemic.

## 1. Introduction

The COVID-19 pandemic was declared by the World Health Organization a Public Health Emergency of International Concern in January 2020, a classification that ended in May 2023. It is among the worst pandemics in known human history and the first to be extensively studied scientifically in all its dimensions [1]. One of the dimensions that COVID-19 substantially impacted was mental health. Mental health reactions are multi-factorial, with the pandemic’s character itself and its emotional consequences playing an important role. Fear of COVID-19 was consistently found to be strongly related to anxiety, traumatic stress and distress, and moderately related to stress and depression [2]. The variety of mental health reactions, the differences in affected groups, and the national differences mean that understanding mental health impacts needs to account for such diversity. Understanding the factors and mechanisms through which the COVID-19 pandemic operates is relevant for policies and interventions addressing mental health for future pandemics.

The wealth of research on the mental health impacts of COVID-19 allows us to draw some conclusions concerning the dimensions that explain the general mental health reaction of a population to a pandemic. The present paper aims to propose a conceptual model of the impacts of this pandemic on mental health. This proposal is theoretically sustained by the possibility of developing theories from data through abductive procedures [3]. In other words, empirical observations can lead to the development of theories as the best explanations for the data. As scientific theories, eventual models that derive from the creative interpretation of data need to be further confronted with new empirical findings in an iterative process. For the present formulation, the empirical foundation is essentially focused on study aggregations. To this end, we reviewed the literature, predominantly relying on systematic reviews and meta-analyses to highlight specific aspects of the model. The search strategy involved exploring all studies intersecting mental health (e.g., specific disorders, mental health symptoms in the community, general stress responses) with COVID-19. We also looked for particular issues related to mental health, such as long-COVID-19, the effects of lockdowns, and impacts on vulnerable populations. This revision, while not exhaustive, aimed at providing a picture of the mental health impacts and their conditions.

By understanding the literature using this conceptual model, we aim to translate the existing and dispersed knowledge. This formulation can influence future policy responses in a scientifically informed manner.

## 2. A Multiaxial Model for Understanding the Impacts of COVID-19 Pandemic

Research findings were organized into axes, making this a multiaxial model. Multiaxial models are familiar in the field of mental health. They served as the means for organizing diagnoses from DSM-III to DSM-IV-RT [4]. They were abandoned in DSM-5 due to coherence problems (e.g., the articulation of affective disorders in axis I and personality disorders in axis II) and the desire to simplify and make diagnoses similar to other medical specialties [5]. The argument for including an axialization is the consideration of multiple dimensions (e.g., temporal, contextual, and functional) in explaining mental health. This need for acknowledging complexity is considered relevant to understanding the mental health impacts of COVID-19.

The proposed axial model is depicted in Figure 1. The first axis refers to the degree of exposure to pandemic-related events. Naturally, this refers to the differential impacts of COVID-19 in different countries or contexts. The second axis relates to vulnerability. This refers to personal vulnerabilities, such as those linked to a previous mental health disorder, or social vulnerabilities, such as being a part of a minority. The third axis is time. This is particularly relevant due to the inherently transient nature of the evolution of the pandemic and human reactions to it. Several psychological and social dimensions have changed over time, and this temporal evolution is crucial in understanding the phenomena. The fourth and final axis refers to external contextual factors, such as economics, healthcare response, and internal mechanisms, for which we focused on meaning-related concepts.

The model is multiaxial to highlight the interactions among these dimensions. A general stress response may be the initial response to the pandemic (3. Time) or the result of the intersection of low vulnerability (2. Vulnerability) with low exposure (1. Exposure). A chronic response could result from a lack of care in an impoverished context or stigma-related beliefs about COVID-19 (e.g., “COVID-infected individuals are to blame for the spread of the infection”). The multiple interactions provided by an axial model allow for integrating some of the complexity recognized in this subject. It is important to consider that the proposed axes are not assumed to be completely independent. For example, the context (i.e., epidemiology of COVID-19, existing social and healthcare support) directly influences exposure. However, by proposing to disentangle the dimensions that explain the effects, we achieve working propositions that can be pragmatically useful and subject to research.

### 2.1. Axis 1: Exposure

Mental health reactions to pandemics are, by definition, reactive. As with most reactive mental health responses, the degree of exposure to the stressful event is relevant. Two observations support this assumption. Firstly, there are specific groups of individuals who, by the nature of their function, are more exposed to COVID-related stressors (e.g., having had COVID-19, witnessing death, and the consequences of this illness). This explains why healthcare workers present higher levels of mental health symptoms [6,7]. The consequences for healthcare workers will depend on the degree of exposure, with frontline healthcare workers experiencing worse reactions. In a meta-analysis of 19 studies, frontline healthcare workers presented greater symptoms of insomnia, stress, anxiety, and depression [8]. This meta-analysis also found regional and temporal differences, which are consistent with varying levels of exposure.

Another relevant group is older adults, who were more affected by COVID-19 in terms of mortality and disease impacts. These impacts are aggravated by specific challenges associated with old age, including negative representations of old age. A systematic literature review [9] showed an increased impact on social and psychological well-being. This includes individuals with dementia and other cognitive limitations. It is also important to remember that this population segment was at a higher risk for COVID-19.

The second observation comes from understanding COVID-19 as a stressor or a set of stressors. In other words, COVID-19 occurs alongside other stressors and can itself be associated with specific stressors (e.g., hospital admission, quarantine, death of relatives). This understanding comes from observing the different impacts of COVID-related events or stressors [10]. Experiencing multiple COVID-related events is associated with stress, and different events have different impacts, with some relevant indirect effects (e.g., tension at home) being particularly stressful. It is important to consider that since this is a cross-sectional correlation study [10], third variables—reflecting pre-existing factors—may help explain some of these effects. In any case, the existence of exposure vulnerability in some groups and the differentiation of events are consonant with the assumption that exposure plays a role.

### 2.2. Axis 2: Personal and Social Vulnerability

Mental health reactions are typically an interaction between vulnerabilities and encounters with stressors [11]. While the general population is expected to experience some reaction, people with specific vulnerabilities may have higher and particular responses. This is the case for people with previous mental health conditions, where relapse or heightened symptoms were observed [12]. It is important to consider that these effects may be disorder-specific. Features of COVID-19 may be related to characteristics of specific mental health disorders: obsessive-compulsive disorder (i.e., via themes of contamination), health-related anxiety (i.e., fear of being infected or dying of COVID-19), depression (i.e., social isolation), etc. Even non-pathological personal transient vulnerabilities are relevant. Several studies found a higher prevalence of mental health issues in mothers during the perinatal period [13]. This can be due to known biological vulnerabilities but also to the specific negative effects of lockdown on family life and parenting. A meta-analysis of 18 studies showed that mothers of young children presented higher mental health symptoms [14] in Europe and North America.

These vulnerabilities need not be understood only individually, as several social vulnerabilities can be observed. Research was conducted on migrants and refugees, particularly those of different ethnic backgrounds. Alarcão and colleagues [15] found specificities in mental health impacts on the mental health of migrants. Analyzing 11 studies in Europe, they found impacts associated with specific mental health factors: economic stability, socio-demographics, lack of healthcare, or adherence. Stigma and family impacts were particularly relevant in these populations. Specific groups such as refugees and asylum seekers are other examples [16] for which the conjugation of general and specific factors will play a role. Among the factors are elements such as racism, which should play a role in the mental health consequences of COVID-19 in general and particular contexts [17]. Among the general factors is the link between negative discrimination and economic hardship, which becomes a vulnerability for mental health reactions.

### 2.3. Axis 3: Time

The need to differentiate several pathological and stress-related reactions also comes from the longitudinal responses to COVID-19. A meta-analysis of 64 longitudinal cases [18] found a decrease in anxiety and depression symptoms during the pandemic but not for other diagnoses (e.g., PTSD, suicidal ideation, and substance use). The peak for depression and anxiety occurred in May, while psychological distress peaked in the months following July 2020. These different temporal patterns suggest qualitatively different phenomena. General symptoms and more frequent diagnoses (such as anxiety or depression) may correspond to extreme cases on a continuum of psychological distress (i.e., from normal to abnormal). Some specific disorders may correspond to psychopathology that developed or re-emerged in stressful conditions. The different temporal evolutions could be a sign of the diverse nature of the reactions.

Another example of the relevance of time for mental health reactions is the consideration of the evolution of COVID-19, which, in some patients, leads to what has been designated as long-COVID-19. Scientific investigations place mental health symptoms as one of the features of this syndrome [19]. These mental health reactions differ from acute mental health reactions.

### 2.4. Axis 4: Context and Social Representations

Not only does a pandemic occur within a context, but that context also plays a specific role in the mental health response. For convenience, we separate the context into external factors and social representations that are shared and are often internalized, mediating the impacts of exposure and emotional/mental health reactions. The context constitutes the background in which the former three axes play out and have multiple interactions with them. Concerning external factors, we will consider three: (1) the economy and social context, (2) healthcare systems, and (3) specific policies.

The link between mental health and poverty or economic hardships is hardly new. However, the economic consequences have been crucial in the context of COVID-19 [20]. There were direct effects of the COVID-19 pandemic and indirect effects via the consequences of lockdown and other protective measures on the economy. These economic effects interacted with and added to the already mentioned dimensions of the pandemic in mental health (e.g., exposure, vulnerability). When comparing the COVID-19 pandemic with the general economic crisis, significant increases in mental health conditions and consequences (e.g., suicides) are found [21]. In other words, the pandemic had a greater impact than an economic crisis due to its cumulative and interactional effects.

The second important factor is a strong healthcare system. Countries with stronger healthcare (more than national wealth) presented better epidemiological outcomes. Concerning the mental health impacts, specific responses were likewise essential. The mental response to the COVID-19 pandemic showed significant variation across countries. A systematic review conducted by Duden et al. [22] reviewed 29 studies analyzing 63 countries’ policy responses to mental health. They found discussions on the lack of preparedness of the systems and their impacts on professionals, as well as the process of changing procedures and practices. Among these adaptations was the use of new technologies to provide services. These adaptations varied across the phases of the pandemic and national contexts. Different governments were responding to the current pandemic [23]. This response was often translated into policies that developed over time. Another adaptation was the use of technology in implementing interventions. The use of technology, especially distance-based online tools, was unique to the response to this pandemic. Most interventions included in these tools had multiple focuses, such as ameliorating symptoms like depression, anxiety, or insomnia [24], and there was evolution over time [25,26] in the proposed interventions. This happened due to a greater awareness of mental health issues and research.

Specific policies also play an important role in the impacts of COVID-19 on mental health. One example of a negative impact was school closures, considered one of the main factors suggested to explain the prevalence of mental health issues in children and adolescents. Harrison et al. [27] found a prevalence of 32% for depression and anxiety in these age groups. These higher prevalences seem to be linked with the impacts of school closures on children’s mental health [28], affecting general distress, anxiety, depression, and specific issues such as inattention and restlessness. Another example is work-related protective measures. Here, in general, little research has been conducted to study the effects on mental health. A scoping of public policies concerning work-related policies for mental health provides a picture of the scarcity of such policies [29]. The attempted strategies involve directly promoting mental health and increasing resources, including addressing self-care problems and reducing treatment barriers. Important contexts, such as work or school, may be subject to pandemic-related policies and interventions. Considering the impacts on mental health and adopting mitigating interventions can be essential to minimizing the impacts on mental health.

However, the societal response seemed to be disproportionally focused on physical health relative to mental health. There was a somewhat haphazard response to mental health problems in general healthcare and public health policies [18]. This is to be expected due to the emergent nature of the response and the need to prevent fatalities. The variation in mental health considerations throughout countries also reflected the adaptation of the response to different national realities. The effectiveness of these multiple mental health policies and interventions is still to be established, and it should vary across different national realities. In any case, the existence of mental health-specific responses should be recommended, given the existing evidence.

The effects of illnesses are not only due to objective factors but also to how they are socially represented. Comparing COVID-19 with another recent epidemic—AIDS—highlights this difference, as the two illnesses are perceived and valued differently. AIDS is a stigmatized condition [30], in a manner that COVID-19 is not. On the other hand, the social response to COVID-19 was influenced by specific social representations. Among these were representations of innocuousness or lack of severity, which were relevant, for example, for beliefs and attitudes about vaccines [31]. Several dimensions of meaning have been linked to COVID-19, such as compared risk, illness perceptions, and self-efficacy [32]. The way COVID-19 is represented is associated with perceived stress, and it has changed throughout the initial stages of the pandemic [33], highlighting the dynamic nature of this process. These meanings may be linked to or explain important outcomes, including adherence to protective behaviors [34,35] and prosocial behavior [36] during COVID-19. Finally, it is important to consider that these social representations are not detached from the general context discussed earlier. Economic crises have a negative effect on the representation of mental health [37], and similar effects should be identified for the COVID-19 pandemic.

## 3. Conclusions and Implications

Understanding the factors associated with mental health problems in COVID-19 is essential to using the wealth of research to inform the response to the next pandemic. We organized the existing research findings into four axes that systematize the current evidence. Some of these propositions are consonant with what is known from general psychopathology—mental health results from the interaction between exposure (axis 1) and vulnerability (axis 2). As a reaction to specific events (i.e., the first case in a given context, the first instance of experiencing COVID-19), mental health responses are temporally bound (axis 3). Finally, COVID-19 affected communities or countries, so the social context is crucial (axis 4). This context includes cultural differences, contextual differences in health provision, public policies, and how these social dimensions are internalized.

The implications for the next pandemic are varied. Two types of implications can be drawn from this model based on each axis and the interactions between them. Firstly, the intervention should be time-sensitive (axis 2) in preparing the mental health-related response. The initial response to a pandemic should focus on general impacts and progressively address more specific impacts or those that imply a more specialized response. Secondly, healthcare and social care (axis 4) are crucial buffers to mental health impacts that mitigate many elements, such as exposure to economic hardships or addressing mental health directly through mental health services. Thirdly, while exposure (axis 1) is difficult to prevent, special care could be given to those more exposed due to their function—e.g., frontline health workers—or due to illness consequences—e.g., interned individuals or grieving relatives. Fourthly, vulnerable individuals should be given special care (axis 3). Individuals with previous mental health disorders, those from social/ethnic minorities, or people facing transient vulnerability factors (e.g., perinatal) should be given greater attention. Finally, while the need to adapt policy to each particular social context (axis 4) should be self-evident, this model suggests that social dimensions are relevant in shaping public perceptions in relevant manners. Different communities perceive an illness differently, affecting public mental health factors.

The second implication is that synergies can be assumed from this model. Public health messages can foster particular meanings that can reduce the stress associated with exposure or for vulnerable individuals (axes 4 and 1 or 2). The model is also helpful in explaining some of the existing observations. As mentioned, anxiety and depression reduced throughout the pandemic, while other symptom reactions (e.g., PTSD and substance abuse) remained high [18]. This difference in reaction could be explained by the interaction between personal vulnerabilities and time (axes 2 and 3). Most individuals reacting in the early stages will not have specific vulnerabilities and may respond through the most common mental health reactions (e.g., distress, anxiety, and depressive symptoms). For those individuals presenting specific mental health reactions, these would more likely correspond to an interaction between personal vulnerabilities and exposure to COVID-related events. As these mental health reactions are more complex, they are expected to be more long-lasting. Another example is the interaction between social representation and exposure (axes 4 and 1). As the consequences for the mental health of AIDS are more problematic in contexts with a higher stigma, the same can be considered in other pandemics, such as COVID-19. The mental health effects of experiencing COVID-19 or protective measures, such as lockdowns and quarantines, may be higher in contexts where the social representations of this illness are more negative.

The organization of the existing evidence in this model can raise several questions. Firstly, the current model is based predominantly on systematic reviews and meta-analyses. This means there were fewer sources of other types of evidence, such as studies on specific topics or qualitative analyses. Such materials could have influenced the model in different directions. This was chosen to provide a broad conceptual framework for the main mental health impacts of a pandemic, rather than focusing on specific or minor impacts. Nonetheless, future research may lead to refinements. Secondly, this formulation was proposed for its pragmatic value in addressing the mental health impact of future pandemics. However, as the comparison with AIDS demonstrates, different illnesses, even with an epidemic character, will show important differences that translate into different forms of mental health impacts. This model may need adaptation for future pandemics. Such adaptations would be informative regarding what is generalizable and what is specific to COVID-19. Thirdly, organizing the factors into axes allows for accounting for different realities (e.g., countries with varying degrees of health systems). However, axes are more suitable when the considered dimensions can be easily differentiated. Some of the elements described in this paper do not differentiate easily, at least quantitatively. Social representations of illness and social vulnerabilities are part of a single broad phenomenon of social representations. Exposure and healthcare realities are intimately interlinked. Some factors, such as old age, are direct risk factors associated with exposure and have personal/social vulnerability dimensions. We assume that the advantages of pragmatic theorization will compensate for these conceptual overlaps.

The mental health response to the COVID-19 pandemic was secondary to the pressing need to save lives. When the impacts became evident, the response was often disorganized or disarticulated. In the next pandemic, the existing evidence allows for informed decisions and actions. The reflection of this paper seeks to enable such actions in a manner that is flexible to contextual realities and the temporal progression of the pandemic.

## Figures and Tables

**Figure 1 behavsci-13-00912-f001:**
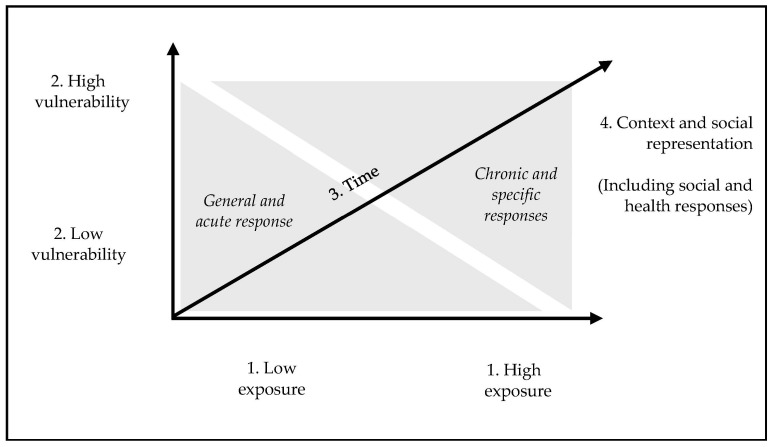
Multiaxial model of pandemic response.

## Data Availability

No new data were created or analyzed in this study. Data sharing is not applicable to this article.

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
