# Peer review of "The Mental Health Impacts of a Pandemic: A Multiaxial Conceptual Model for COVID-19"

_behavsci, 2023, doi:10.3390/bs13110912_

Round 1

Reviewer 1 Report

Comments and Suggestions for Authors

The manuscript with the title “The mental health impacts of a pandemic: A multiaxial conceptual model for COVID-19” (behavsci-2643712) is quite well written. 

The multiaxial model is a suitable framework to integrate former results towards Covid & mental health. Is this model an adaptation from other authors?

In the following I have some further comments / suggestions:

 Line 28: “It was one of the deadliest pandemics in known human history” -> which citation supports this?

 Line 77/78: “(…) or a self-stigma belief about COVID” -> what do you mean exactly? Can you give an example?

 Line 99 pp: old age to me seems more suitable as part of Axis 2 Personal and Social Vulnerability

 Line 105 pp: “The second observation comes from understanding COVID as a stressor or a set of

stressors. This understanding comes from observing the different impacts of COVID-re-

lated events or stressors” -> I think it would be more understandable for the readers if this could be further carried out

 Line 110: which studies are cross-sectional correlation studies?

 Line 120 pp: I think mental disorders can be carried out more sophisticated and multifaceted: 1) obsessive-compulsive disorder are not necessary linked to contamination, its contents can relate to anything. 2) depression and anxiety disorders are clearly linked to Covid (e.g. less social contacts, less structure of everyday life, fear of infection…), 3) what about social phobia for instance? Those people even had a benefit through avoidance of social contacts (which of course can only be beneficial in the short-term but not in long-term).

 Line 135: racism and Covid? I do not see the connection

 Line 144 pp: “These results suggest that general psychological distress and mental health disorders may follow a different pattern, indicating that they may be qualitatively different phenomena.” -> how are these phenomena specifically different to each other? [also Line 151]

 -> methods are lacking overall: which  literature databases were used? Which keywords for literature search did you use?  How was the framework (multiaxial conceptual model) developed?

 -> conclusions and implications: in my opinion the contextual aspects (axis 4) were not taken into account sufficiently. Measures conducted by different countries (social distancing etc.), wearing masks and especially vaccinations showed different effects on the population. While some felt protected or relief in terms of fears, others felt constricted with different effects on mental health. These different effects were not included in an appropriate way. Also, implications are rather superficial and it would be helpful to be more specific where possible.

Author Response

We would like to express our gratitude to reviewer 1 for her/his comments.

We have increased the explanation on the basis of the model in the penultimate paragraph of the introduction and the first paragraph of the second section (page 2). We have also added two references: Haig (2005) and Wakefield (2013)

This model is the application of general psychopathology to COVID-19 as it is recognized in the beginning

In the following I have some further comments / suggestions:

 Line 28: “It was one of the deadliest pandemics in known human history” -> which citation supports this?

According to estimates, it was the 5th deadliest: 1) Black Death  - 75–200 million; 2) Spanish flu - 17–100 million; 3) Plague of Justinian 15–100 million; ) HIV/AIDS - 42 million; 5) COVID-19 - 6.9–31.4 million (Wikipedia)

Considering that data for the deadliest pandemics is very imprecise, we just added a reference with the estimation of the number of casualties for COVID-19. We also toned down the sentence

 Line 77/78: “(…) or a self-stigma belief about COVID” -> what do you mean exactly? Can you give an example?

We generalized to general stigma by removing “self” and added an example as requested.

 Line 99 pp: old age to me seems more suitable as part of Axis 2 Personal and Social Vulnerability

We agree that old age could be seen as a personal/social vulnerability. We included it within the first axis, considering that old age was a specific risk factor for COVID itself – rather than only the mental health impacts.

Due to the biological interaction between the virus and immune response in older individuals, the ACTUAL severity/risk was higher. The decision to include in the first axis was due to these characteristics.  We do, however, acknowledge that for mental health impacts, old age was a factor.

We have added a sentence in the limitations (the penultimate paragraph of the paper) to use age as an example of issues that may overlap between axes

 Line 105 pp: “The second observation comes from understanding COVID as a stressor or a set of stressors. This understanding comes from observing the different impacts of COVID-related events or stressors” -> I think it would be more understandable for the readers if this could be further carried out

We have added a sentence.

Line 110: which studies are cross-sectional correlation studies?

We have corrected the sentence. We hope to have made it clearer.

Line 120 pp: I think mental disorders can be carried out more sophisticated and multifaceted: 1) obsessive-compulsive disorder are not necessary linked to contamination, its contents can relate to anything. 2) depression and anxiety disorders are clearly linked to Covid (e.g. less social contacts, less structure of everyday life, fear of infection…), 3) what about social phobia for instance? Those people even had a benefit through avoidance of social contacts (which of course can only be beneficial in the short-term but not in long-term).

Thank you to pointing this out. We have reformulated accordingly.

Line 135: racism and Covid? I do not see the connection

There is a reference to support such a link. Considering that the virus was initially addressed as Chinese flu – migrants were initially more stigmatized (from migrant internal workers in India to foreign migrants in other contexts).

https://www.nytimes.com/2022/02/25/health/covid-racial-ethnic-discrimination.html

Line 144 pp: “These results suggest that general psychological distress and mental health disorders may follow a different pattern, indicating that they may be qualitatively different phenomena.” -> how are these phenomena specifically different to each other? [also Line 151]

Thank you for calling for this need for clarification. We have reformulated the segment.

-> methods are lacking overall: which  literature databases were used? Which keywords for literature search did you use?  How was the framework (multiaxial conceptual model) developed?

- We have expanded substantially on page 2 of the strategy and what fundaments the development of the model.

-This is not a systematic review but rather a reflection on the existing literature based on aggregate studies – meta-analysis and systematic reviews. We were looking for a general framework for understanding broad mental health impacts. Considering that this could be questioned, we added it as a contextualization in the conclusion (page 7)

- Theory building in science is assumed to have a certain creative element. This, like any other formulation, should then be subjected to critical appraisal against new or existing empirical data.

-> conclusions and implications: in my opinion the contextual aspects (axis 4) were not taken into account sufficiently. Measures conducted by different countries (social distancing etc.), wearing masks and especially vaccinations showed different effects on the population. While some felt protected or relief in terms of fears, others felt constricted with different effects on mental health. These different effects were not included in an appropriate way. Also, implications are rather superficial and it would be helpful to be more specific where possible.

We agree with the point made by Reviewer 1. In fact Axis 4 was the one with the most literature review in the sections describing the axis framework.

We have expanded the exploration of the implications of contextual aspects in the conclusions – but perhaps it will still seem insufficient as perhaps an entire paper could be written just on this dimension.

-> We wish to thank Reviewer 1 for its exhaustive and critical reading. We believe that these comments have enriched the paper.

Reviewer 2 Report

Comments and Suggestions for Authors

The authors through this article aim to synthesize the impact of the pandemic on mental health through a multiaxial conceptual model.

Although the attempt made is very appreciable the methodological aspects are very unclear. How is the multiaxial conceptual model carried out? By what criterion were some studies considered and not others? How was the definition of these four axes arrived at? 

Since the methodological procedures have not been clarified everything appears extremely arbitrary.

Therefore, it is recommended to add a paragraph with the methodological aspects related to the multiaxial conceptual model, as well as data on the articles considered, their eligibility criteria, and the criteria that led to the definition of the axes.

The second paragraph is redundant to what has been reported in more detail in the subsections describing each axis.

Author Response

The authors through this article aim to synthesize the impact of the pandemic on mental health through a multiaxial conceptual model.

Although the attempt made is very appreciable the methodological aspects are very unclear. How is the multiaxial conceptual model carried out? By what criterion were some studies considered and not others? How was the definition of these four axes arrived at? 

Since the methodological procedures have not been clarified everything appears extremely arbitrary.

Therefore, it is recommended to add a paragraph with the methodological aspects related to the multiaxial conceptual model, as well as data on the articles considered, their eligibility criteria, and the criteria that led to the definition of the axes.

-> We would like to thank Reviewer 2 for the time spent in reading our paper and her/his comments.

-> We agree with the reviewer that the model itself is a perspective or theorization of the data. As such, there is an interpretative or creative dimension to it – as it happens in other models or theories in science. To meet the reviewer's concern, we included a few alterations: a) we included in the introduction a reference (Haig, 2005) to the philosophical underpinning of our theory-building; b) in the introduction, we expanded the description of our literature search and revision; c) we added in the conclusions section a note of caution as to the possible consequences of overlooking specific studies or types of data.

The second paragraph is redundant to what has been reported in more detail in the subsections describing each axis.

.> We agree with this appreciation. We have deleted the paragraph and moved some of its segments to the corresponding sub-sections when appropriate.

Reviewer 3 Report

Comments and Suggestions for Authors

The multi-axial model they propose is very illuminating and brings together the different approaches that researchers have been and are still addressing.
The references on which they base their arguments to make sense of the content of the axes are based on robust evidence.
Although it is true that the transfer of this knowledge that they gather and order according to these axes remains quite open and perhaps not sufficiently concrete.
However, it is true that, as the authors themselves point out, the particularities of each pandemic prevent the protocols to be implemented from being more specific.
The article is enlightening in bringing together information and conclusions in a common model.
I would only briefly suggest two aspects:
- Explain how you are going to proceed in order to provide the information and evidence for the proposed axes. In other words, between the introduction (point 1) and the explanation of the model (point 2), include a paragraph explaining the procedure of the reviews carried out: criteria, number of works included and excluded, etc.
- Clarify a little more the conclusions on the impact on Mental Health that you state in the section on conclusions and implications.
Congratulations for your effort and contributions

Author Response

The multi-axial model they propose is very illuminating and brings together the different approaches that researchers have been and are still addressing.
The references on which they base their arguments to make sense of the content of the axes are based on robust evidence.
Although it is true that the transfer of this knowledge that they gather and order according to these axes remains quite open and perhaps not sufficiently concrete.
However, it is true that, as the authors themselves point out, the particularities of each pandemic prevent the protocols to be implemented from being more specific.
The article is enlightening in bringing together information and conclusions in a common model.
I would only briefly suggest two aspects:
- Explain how you are going to proceed in order to provide the information and evidence for the proposed axes. In other words, between the introduction (point 1) and the explanation of the model (point 2), include a paragraph explaining the procedure of the reviews carried out: criteria, number of works included and excluded, etc.

-> We want to convey our gratitude to Reviewer’s 3 reading of our paper and her/his comments and suggestions. We believe they have enriched our paper.

-> We added on page two, before section 2 further details relative to the literature search strategy. We focused on systematic reviews and metanalysis – as aggregates of the literature. However, this is not a systematic review, but rather a review of the existing literature to reflect and organise the factors that are associated with mental health issues. As with any other theorization in a scientific discipline, it has some interpretative or creative dimension. Future evidence will help us question and reformulate this theorization.

- Clarify a little more the conclusions on the impact on Mental Health that you state in the section on conclusions and implications.
Congratulations for your effort and contributions

-> We expanded the conclusions and clarified them in terms of the axes proposed in the model. We hope to have met the reviewer’s concerns.

-> We want to convey our gratitude to the Reviewer’s 3 readings of our paper and her/his comments and suggestions. We believe they have enriched our paper.

Round 2

Reviewer 1 Report

Comments and Suggestions for Authors

Dear authors, thank you for your decisive revision, which improves the manuscript and makes it clearer overall.

Reviewer 2 Report

Comments and Suggestions for Authors

Dear Authors,

I appreciated the revision work you conducted on the article. I believe that the most critical points have been adequately clarified.